# Factors Associated with Symptom Burden in Adults with Chronic Kidney Disease Undergoing Hemodialysis: A Prospective Study

**DOI:** 10.3390/ijerph19095540

**Published:** 2022-05-03

**Authors:** Thalwaththe Gedara Nadeeka Shayamalie Gunarathne, Li Yoong Tang, Soo Kun Lim, Nishantha Nanayakkara, Hewaratne Dassanayakege Wimala Thushari Damayanthi, Khatijah L. Abdullah

**Affiliations:** 1Department of Nursing Science, Faculty of Medicine, University of Malaya, Kuala Lumpur 50603, Malaysia; mva180022@siswa.um.edu.my or; 2Department of Nursing, Faculty of Allied Health Sciences, University of Peradeniya, Peradeniya 20400, Sri Lanka; damayanthi74@ahs.pdn.ac.lk; 3Department of Medicine, Faculty of Medicine, University of Malaya, Kuala Lumpur 50603, Malaysia; limsk@ummc.edu.my; 4Nephrology and Dialysis Unit, National Hospital, Kandy 20000, Sri Lanka; nishantha4313@gmail.com; 5Department of Nursing, School of Medical and Life Science, Sunway University, Bandar Sunway 47500, Malaysia; khatijahl@sunway.edu.my

**Keywords:** symptom burden, adults, hemodialysis, Sri Lanka, longitudinal study

## Abstract

People with end stage renal disease and undergoing hemodialysis experience a high symptom burden that impairs quality of life. This study aimed to assess the prevalence, dynamicity and determinants of symptom burden among middle-aged and older adult hemodialysis patients. A descriptive cross-sectional study together with a longitudinal assessment was used. A total of 118 and 102 hemodialysis patients were assessed at baseline and at a 6-month follow-up. Validated questionnaires were used to assess the symptom burden, stress, illness perception and social support. Multiple linear regression analysis was used to determine the factors associated with symptom burden. The median number of symptoms experienced was 21 (Interquartile Range (IQR); 18–23) and 19 (IQR; 13–22) at baseline and 6 months, respectively. Having elevated stress (β = 0.65, *p* ≤ 0.005) and illness perception (β = 0.21, *p* = 0.02) were significantly predicted symptom burden at baseline (F (4, 112) = 55.29, *p* < 0.005, R^2^ = 0.664). Stress (β = 0.28, *p* = 0.003), illness perception (β = 0.2, *p* = 0.03), poor social support (β = −0.22, *p* = 0.01) and low body weight (β = −0.19, *p* = 0.03) were the determinants for symptom burden at 6 months (F (5, 93) = 4.85, *p* ≤ 0.005, R^2^ = 0.24). Elevated stress, illness perception level, poor social support and low post-dialysis body weight were found to be determinants for symptom burden. Attention should be given to psychosocial factors of hemodialysis patients while conducting assessment and delivering care to patients.

## 1. Introduction

The world’s population is aging. Sri Lanka has been identified as one of the fastest aging countries in South Asia due to rapid economic growth which has, in turn, improved its health indicators [1]. The report of the World Population Prospectus 2019 [2] showed that the percentage of the population aged above 65 years in Sri Lanka is expected to undergo a rise of 21% by 2045 and of 35.6% by 2100 [2]. This rapid demographic transition will cause detrimental effects to older people resulting in ill health and frailty, as well as creating a need for long-term care due to non-communicable diseases (NCDs). Chronic kidney disease (CKD) has become a leading NCD and is ranked as the 12th most common cause of death [3]. Disregarding the underlying cause of CKD, a substantial proportion of middle-aged and older adult people progress to end-stage renal disease (ESRD) requiring dialysis therapy [4,5]. In Sri Lanka, CKD and its main treatment modality, hemodialysis, is experiencing increasing demand due to the rapid increase in the prevalence of CKD and the emergence of CKD of unknown origin (CKDu) [6]. Occurrence and diagnosis of CKD is predominantly among male farmers aged above 40 years [7,8,9,10]. Thus, according to the report of the World Health Organization, CKD is primarily found in the middle-aged and older adult population [11].

Patients with ESRD exhibit symptoms in clusters rather than in isolation [12]. CKD patients often report physical and psychological symptoms such as fatigue, anorexia, pain, nausea, pruritis, shortness of breath, muscle cramps, paresthesia, depression, sexual difficulty and sleep disturbance [13,14,15]. The quantitative representation of frequency and severity of perceived symptoms is referred to as the “symptom burden” [16].

The theory of unpleasant symptoms describes the experience of symptoms and multidimensional factors affecting symptom perception [17]. According to the model, factors influencing the perceived symptoms among hemodialysis patients are threefold: physiological, psychological and situational. However, the relationships between physiological [12,18] (clinical parameters), situational [12,19,20] (demographic and social support) factors and symptom burden has been extensively investigated. Little was known about the association between psychological factors and symptom burden. As an example, increased symptom burden was found to be correlated with depression [12,21]. Nevertheless, the influence of stress and illness perception on symptom burden is poorly documented.

Symptom burden was recorded and quantified in diverse chronic disease populations such as diabetes mellitus [22], heart disease [23,24], renal failure [25,26] and even hemodialysis [27,28]. However, only a few cross-sectional surveys have been conducted to assess symptom burden among the CKD or hemodialysis population in Sri Lanka. One of those studies reported moderate to severe physical symptom burden among CKD patients undergoing hemodialysis in Sri Lanka [29]. Another study found that patients at all stages of CKD experienced high symptom burden with an elevated fatigue level [30]. Moreover, they found that determinants for symptom burden of CKD patients included being educated up to advanced level, having CKD stage V, being dialyzed, having comorbidities and being employed.

However, the absence of any comprehensive investigation into the determinants of symptom burden while patients follow their dialysis program indicates that little attention is being paid to patients’ perspectives on their physical and psychological suffering. Importantly, a hospital-based Sri Lankan study has found that the majority of middle-aged and older adults with ESRD have undergone hemodialysis therapy [5]. Unfortunately, little was known at the time about symptom prevalence, dynamics of symptom burden and factors associated with symptom burden while receiving regular hemodialysis in this cohort, and many studies to date have been restricted to cross sectional discipline rather than longitudinal follow-up. Therefore, this study aimed to assess the prevalence, dynamicity and determinants of symptom burden among middle-aged and older adult patients on hemodialysis.

## 2. Materials and Methods

### 2.1. Study Design, Participants and Setting

A descriptive cross-sectional study, together with a longitudinal assessment, was used. Patients of 40 years and above were recruited because of the high prevalence of CKD [31,32] and possible age-related renal subclinical changes [33]. Since the symptom burden showed an association with a longer duration of dialysis, the newly hemodialysis-initiated patients who were at the beginning of their dialysis program (less than 3 months) were considered in this study [12,34]. Further, participants were recruited if they were receiving daytime hemodialysis with a frequency of 3 times a week, and the ability to understand, communicate in and write Sinhala or Tamil. Nocturnal hemodialysis patients as recorded in the dialysis registry, those with language barriers and those with any acute or chronic disease conditions identified at entry assessment were excluded. The participants were recruited from July to September 2019 and the follow up was conducted after 6 months to baseline.

Participants of this study were registered as regular hemodialysis patients of the Nephrology and Transplant Unit of the National Hospital, Kandy, Sri Lanka. This is the largest renal care service in the country and a referral center for dialysis for local CKD patients including those from CKD hotspots, as it is located in Central Province, Sri Lanka. Three hemodialysis units: Unit I, Unit II and the Emergency Dialysis Unit were functioning at the time of data collection. All three units are capable of carrying out 70 hemodialysis sessions during daytime. 

### 2.2. Sample Size

A total of 520 hemodialysis patients were registered in the clinic. Among them, 118 were aged over 40 years and at the beginning (less than 3 months) of the dialysis program. Thus, all of them were recruited to the baseline assessment (Figure 1). Out of the 118 patients recruited for the baseline assessment, 16 patients were lost by follow up (seven went for a kidney transplant and nine died); thus, a total of 102 patients were assessed in the 6 months.

### 2.3. Measures

#### 2.3.1. Symptom Burden

At baseline and at the 6-month follow-up, the hemodialysis patients received a questionnaire with Chronic Kidney Disease Symptom Index–Sri Lanka (CKDSI-SL) scale to assess their symptom prevalence, symptom severity and symptom burden [30]. This consists of a checklist of 25 symptoms that requires a response of ‘Yes’ or ‘No’. Then, the severity of the symptoms experienced is rated on a five-point Likert scale rated from 1–5 with the following response options: ‘very mild’, ‘mild’, ‘moderate’, ‘severe’ and ‘very severe’. For the symptoms of not experienced, ‘0’ is rated, corresponding to the response option of ‘No’. Thus, the total score can be calculated with the sum of all the individual scores, ranging from 0 to 125 [30]. A higher score reflects a high symptom burden among adult hemodialysis patients. The developers of CKDSI-SL also proved high convergent validity (*p* < 0.001), discriminant validity (*p* < 0.001) and test re-test reliability (Spearman’s r > 0.9) [30]. The translated and validated Sinhala and Tamil version of CKDSI-SL was used in this study.

#### 2.3.2. Socio-Demographic Characteristics

The following socio-demographic characteristics were collected: age, gender, marital status (unmarried and ever married), education level (primary, secondary and tertiary), occupation (government, private, unemployed and self-employed), average monthly income, family history of CKD, history of diabetes, hypertension, acute kidney injury, coronary artery disease, stroke and sedentary lifestyle.

#### 2.3.3. Anthropometric Measurements and Clinical Characteristics

Basic anthropometric measurements, such as body weight and height were measured. Body weight was measured in kilograms before the commencement (pre-dialysis) and at the end of the hemodialysis procedure (post dialysis). During the measurement, patients were required to wear lightweight clothing and no shoes. The body mass index (BMI) was calculated using weight in kilograms (kg) divided by height in meters (m) squared. The standard procedures such as triplication and mean value of weight and height measurement was used to calculate BMI. Post dialysis body weight was taken as the weight measurement [35] in calculating patients’ BMI. The following values were used to determine the BMI categories of adults starting dialysis [36].

Lower BMI (<23.1 kg/m^2^)

Normal BMI (23.1–26.0 kg/m^2^)

Higher BMI (≥26.0 kg/m^2^)

The most recent value of hemoglobin (g/dl), serum albumin (mg/dl), serum creatinine (mg/dl), blood urea nitrogen (BUN) (mg/dl), and Kt/V were obtained from the patients’ dialysis records as clinical parameters during baseline and the 6-month assessment.

#### 2.3.4. Stress

The translated Perceived Stress Scale-10 (PSS-10) was used to assess stress level. It had 10 statements to respond to on the Likert scale, rated from 0–4. Thus, for each statement patients were asked to choose their answer from the following alternatives: ‘never’, ‘almost never’, ‘some times’, ‘fairly often’ and ‘very often’. Reverse responses of Questions 4, 5, 7 and 8 were added to the scores of the other questions to generate the total PSS score. The total score of the PSS-10 ranges from 0 to 40 with higher scores indicating higher perceived stress [37]. Cronbach’s alpha for PSS-10 was 0.82.

#### 2.3.5. Illness Perception

The nine-item Brief Illness Perception Questionnaire (BIPQ) was used to assess patients’ cognitive and emotional representations related to their illness [38] and recorded on a continuous linear scale. The first five components in the BIPQ (consequences, timeline, personal control, treatment control and identity) assess patients’ cognitive status. Concern and emotions measure the emotional status of the patients. The illness comprehensibility was also recorded. Lastly, an open-ended question assessed the verbal causal explanations [38]. The total illness perception score indicated the patient’s extent of illness threat, and the total score was calculated as follows: the reverse-scores of items 3, 4 and 7 were summated and added to the score of items 1, 2, 5, 6 and 8. Thus, the total possible score could range from 0 to 80 [38]. Higher BIPQ scores indicate a threatening view of illness. Cronbach’s alpha for BIPQ in this study was 0.75.

#### 2.3.6. Social Support

The translated, culturally validated Sinhala and Tamil versions of Social Support Questionnaire-6 (SSQ6) [39] were used to assess the perceived social support and satisfaction with said social support. Each question of SSQ6 had two parts. The first part asked the participants to name their foremost support individuals, and the second part asked them to rank their level of satisfaction with this support on a 6-point Likert scale. Response options included ‘very dissatisfied’, ‘dissatisfied’, ‘moderately satisfied’, to ‘satisfied’, ‘very satisfied’ and ‘extremely satisfied’. Each item yielded a satisfaction score between 1 and 6, hence the minimum total was 6, and the maximum total, 36 [39]. Cronbach’s alpha for SSQ6 in this study was 0.78. A total score above the 75th percentile is considered as having good social support.

### 2.4. Ethical Considerations

This research was conducted in accordance with the Helsinki statement regarding human research. Ethical approval was taken by the Ethics Review Committee of the Faculty of Allied Health Sciences, University of Peradeniya (Number: AHS/ERC/2019/026). Informed written consent was taken from each participant before the baseline visit and before any procedure was undertaken.

### 2.5. Statistics

The Kolmogorov–Smirnov test was performed to explore the normality. The results of the descriptive analysis were reported with means (standard deviation). Comparison of mean score of anthropometric, clinical parameters, psychosocial factors and symptom burden between two age groups at baseline and the 6-month follow-up was computed using an independent sample *t*-test or Mann–Whitney U test. Univariate analysis was performed separately for baseline and the 6-month follow-up. Depending on the normal/skewed distribution of the data set, an independent sample *t*-test for continuous variables or the Mann–Whitney U test/Kruskal-Wallis H test for categorical variables were used to compare the two age groups. Statistically significant independent variables identified in the bivariate analysis were included in the multiple linear regression model to identify the determinants of symptom burden at baseline and 6 months. A ‘*p*’ value of less than 0.05 was considered statistically significant. The assumptions for regression analysis were tested to ensure the normality, linearity and multicollinearity. All data (Appendix A) were analyzed with IBM Statistical Program for Social Sciences (SPSS) Version 25.

## 3. Results

### 3.1. Patient Characteristics

Table 1 shows the demographic characteristics of the study sample at baseline and at the 6-month follow-up.

At baseline, the sample consisted of 75 males and 43 females. The following two age categories were defined: 40–55 years, and 56 years and above [32,40,41,42]. The majority of patients (44%) were represented in the 40–55 age category. Among the total participants, 96% were, or were once married, which encompassed being married, divorced or widowed. The majority of participants were educated up to secondary level (56.8%), were unemployed (45.8%) and had an income above 25,000 LKR per month (46.6%). The most common co-morbidity was hypertension (38.1%) and 39 patients had a history of both diabetes and hypertension.

Table 2 shows mean (SD) values of anthropometric, clinical parameters, psychosocial factors and symptom burden of patients. A significant difference in Hb% was observed between the two age categories at baseline. However, in the 6-month assessment, only MpreBP showed a significant difference among the studied age groups.

The calculated blood pressure measurements: MpreBP and Mpost BP of both age groups were within the normal ranges. Patients of both age groups at baseline had a low BMI, whereas only those aged between 40–55 years reported a higher BMI in the 6-month assessment. There were no significant differences in mean total scores of stress, illness perception, social support and symptom burden among the two age groups.

### 3.2. Prevalence, and Burden of Symptoms

The median number of symptoms experienced by the hemodialysis patients was 21 at baseline (IQR; 18–23). At the 6-month follow-up, the median number of perceived symptoms was reduced to 19 (IQR; 13–22). Table 3 displays the prevalence of most common symptoms of both age groups at baseline and at 6 months. The most prevalent symptoms of baseline were fatigue (97.5%), decreased libido (90.7%) and loss of appetite (91.5%).

Other than fatigue and loss of appetite, insomnia was identified as the second most prevailing symptom at 6 months. The least prevalent symptoms of baseline were heart burn (55.1%) and loss of memory (51.5%) and at 6 months were hiccups (33.3%) and diarrhea (37.3%), respectively. There was no significant difference in the prevalence of any symptom between the two age groups at baseline. However, a significant difference in prevalence among the two age groups was identified for decreased libido (*p* = 0.02) and impotence (*p* = 0.02) at 6 months.

### 3.3. Factors Associated with Symptom Burden among Patients on Hemodialysis

The results of the univariate analysis are summarized in Table 4. In the univariate analysis, among all studied socio-demographic variables, only having comorbidities showed a positive correlation with symptom burden at baseline (X^2^(3) = 8.6, *p* = 0.03) and at 6 months (X^2^(3) = 9.5, *p* = 0.02).

Among studied anthropometric and clinical parameters, the body weight (r = −0.22; (0.03) had a negative correlation with symptom burden. A strong positive correlation was noted between the total symptom burden score and stress (r = 0.8; 0.005) and illness perception (r = 0.7; 0.005) at baseline. Moreover, the correlational analysis of 6 months found that the total symptom burden score was associated with stress (r = 0.34; 0.005), illness perception (r = 0.24; 0.01) and social support (r = −0.2; 0.04).

Table 5 shows the results of the regression models developed through multiple linear regression. Having elevated stress (β = 0.65, *p*= <0.005) and illness perception level (β = 0.21, *p* = 0.02) were strongly associated with symptom burden at baseline. The total variance demonstrated in the model at baseline was 66.4% (F (4, 112) = 55.29, *p* < 0.0005). The multiple regression analysis of 6 months showed that 24.3% of the variance in symptom burden can be accounted for by four determinants (F (5, 93) = 4.85, *p* = <0.005). The determinants were high stress (β = 0.28, *p* = 0.003) and illness perception level (β = 0.2, *p* = 0.03), poor social support (β = −0.22, *p* = 0.01) and low body weight (post dialysis weight) (β = −0.2, *p* = 0.03).

## 4. Discussion

This study examined the dynamicity of symptom burden and factors associated with symptom burden among middle-aged and older adult patients undergoing hemodialysis. Since the present study is the first ever investigation to assess the dynamics of symptom burden among this cohort in Sri Lanka, findings will be beneficial to patients, care givers and health care workers to manage symptoms promptly. Male preponderance was observed in both study phases of our study, as in other studies [29,43]. More than 44% and 37% of patients at baseline and at six months were aged between 40–55 years, similar to previous findings [43]. Although the most common co-morbidity was hypertension at baseline (38.1%) and at six months (39.2%), more than 33% of patients had multi co-morbidity status (diabetes and hypertension), which is consistent with previous findings [44]. Considering the anthropometric data, the results of the present study found that only post-dialysis body weight showed an inverse association with symptom burden. Moreover, mean post dialysis body weight was reduced in the follow up compared to baseline. A similar finding was reported in the literature [34]. The majority of studies have considered BMI as a study variable rather than post-dialysis body weight (dry weight). Although BMI is not a significant factor for symptom burden in our study, the mean BMI values at baseline (21.86 ± 4.5) and at six months (21.33 ± 4.6) were less than the standard lower BMI value (<23.1 kg/m^2^). Having a lower BMI value was significant among CKD [45,46] and hemodialysis [36] patients and it has been found to be associated with a high risk of mortality and morbidities among patients starting hemodialysis [47,48].

Several important findings were made from this study about symptoms and symptom burden at the start of hemodialysis. With regard to the symptoms, middle-aged and older adult patients on hemodialysis experienced 21 different symptoms at the start point of the hemodialysis schedule, and after six months this number was reduced to 19. Our findings were in line with a previous community-based study conducted in Anuradhapura district, Sri Lanka [49]. They also found that, of the inquired 25 symptoms, 22 were more prevalent among the older age group [49]. In contrast, a recent study conducted in a CKD endemic area found that most patients experienced one to nine symptoms [45]. We inquired about 25 symptoms, and among them, the most commonly reported symptoms at baseline were fatigue, decreased libido, loss of appetite, swelling and dry skin. Fatigue remained the most common symptom at six months, followed by insomnia, loss of appetite, difficulty in keeping legs still and sadness. Similar findings were reported by the cross-sectional study that was conducted using hemodialysis patients from 13 nephrology units of Sri Lanka. The most commonly reported symptoms were tiredness and lack of energy (73.33%), shortness of breath (65.95%), leg swelling (56.22%) and muscle cramps (53.05%) [29]. Yet another study found that bone/joint pain (87.6%), feeling irritable (78.6%) and muscle cramps (77.5%) were common symptoms [30].

A total of 118 and 102 hemodialysis patients were evaluated at baseline and at the six-month assessment, respectively, and at the baseline phase they had moderate symptom burden with a mean of 67.04 ± 22.6. Although the moderate symptom burden persisted even after six months from the initiation of hemodialysis, the mean total symptom burden score fell to 53.1 ± 29.7. This finding bears similarity to a longitudinal study conducted in Australia, where researchers found that symptom burden of hemodialysis patients declined six months from the start of hemodialysis compared to the non-dialysis patients which had higher symptom burden six months from the start of hemodialysis [50]. Importantly, another recent longitudinal study on symptom burden of hemodialysis patients demonstrated that a dramatic improvement in symptom burden among patients over a short follow-up period of three months [51]. Both of these studies bear a similarity to our own finding, that is, a decrease in the total symptom burden among our patients by the follow up. Another two Sri Lankan studies also reported on moderate and severe symptom burden among CKD patients [29,30].

Although numerous studies in the literature have focused on symptom burden among CKD patients, the assessment of symptom burden and associated factors to symptom burden among middle-aged and older adult patients undergoing hemodialysis has been poorly investigated. Stress, illness perception, poor social support and low body weight were the major determinants of a deterioration in the reported symptoms over the six months till follow-up.

Stress was a determinant for perceived symptom burden at the beginning as well as over the six months till follow up. This finding is consistent with another study [52], in which most patients were stressed and reported high symptom burden. Stress has also been identified as a determinant for CKD, as well as having implications for CKD, such as symptom burden [53]. A recent survey in Sri Lanka found that 75% of CKD patients were psychologically distressed and they perceived high symptom burden [30,49], which is attributed to the additional burden of CKD in certain geographical locations in Sri Lanka [54].

Our findings showed that illness perception is also a determinant of symptom burden at both assessment phases. This result is in line with previous research findings [55,56,57]. Illness perception is an organized set of thoughts and beliefs of an individual, generated in response to a health threat. These illness perceptions are triggered by our understanding of the condition, and past and current experience of symptoms related to the condition [58]. These negative experiences have a cognitive and emotional effect and conceptualize the symptoms related to CKD [59]. The hemodialysis patients of our study had a high illness perception score (that is, many had a lot of negative thoughts about the CKD) at baseline and this further increased at the follow up. This suggests that as CKD progresses, and the more dialysis cycles that are received, the middle-aged and older adult hemodialysis patients develop negative thoughts toward CKD. Hence, it is advised that interventions are delivered to reduce illness perception at an early stage, before beliefs have become fully established [60].

Poor social support is another factor associated with symptom burden in our study. This finding is supported by the outcome of a recent study in which the researchers highlighted the importance of social support as a determinant for symptom burden among hemodialysis patients [61]. The plausible explanation for this association is that the presence of adequate social support would be beneficial in overcoming the economic burden as well as relieving psychological stress, thereby reducing the symptom burden. Moreover, patients with chronic disease conditions would experience psychological dependence resulting in depressive symptoms [62]. A recent study conducted among hemodialysis patients found that a good network of social support reduced the depressive symptom burden [63].

Several limitations were encountered in this study. A high drop-out rate was reported in this study. About 14% of patients were lost to the six-month follow-up while they were on the regular hemodialysis schedule. This study was conducted in a single hemodialysis center, thus it reduces the ability to generalize the results. Although this study was conducted as a longitudinal study, the cross-sectional assessment of factors cannot provide direct cause-and-effect associations. The study variables such as stress, illness perception and perceived social support are more likely to be subjective data. Although we quantified answers using a scale, a better expression of the subjective nature of these variables could be extracted via a qualitative study.

## 5. Conclusions

Our findings suggest that middle-aged and older adult patients undergoing hemodialysis experience a high symptom burden at the outset, but a slight reduction was noted at the six-month follow-up. Health care providers should pay rigorous attention to identifying patients with elevated stress and illness perception levels as they appear to negatively affect the symptom burden, both at the start of hemodialysis and even after six months into the schedule. Furthermore, an assessment of available social support is needed, as poor social support is a determinant of symptom burden, especially as hemodialysis advances. In terms of anthropometric measurements, post-dialysis body weight was significantly associated with symptom burden, thus the accurate measurement and documentation of post-dialysis body weight should be encouraged.

## Figures and Tables

**Figure 1 ijerph-19-05540-f001:**
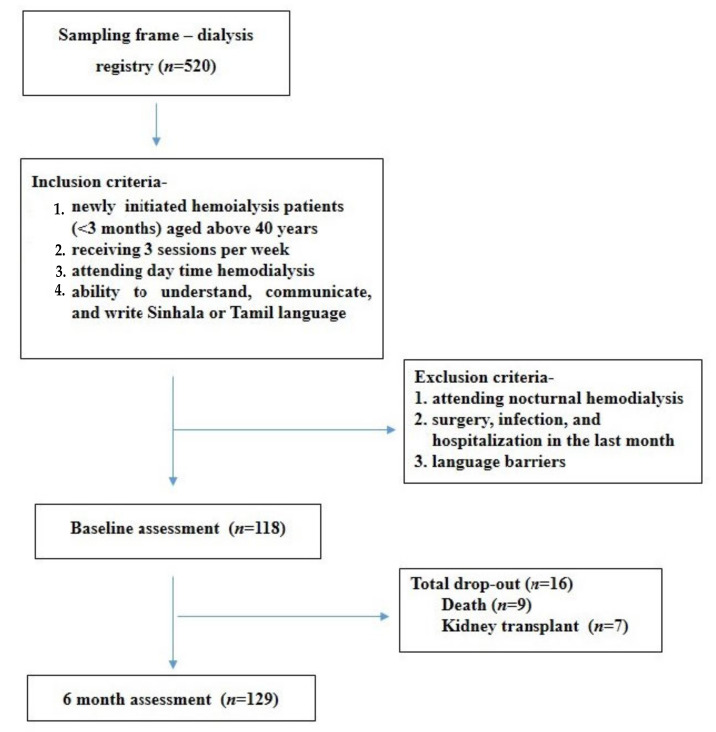
Patient recruitment procedure for the study.

**Table 1 ijerph-19-05540-t001:** Socio-demographic characteristics of study participants at baseline (*n* = 118) and at 6 months (*n* = 102).

Characteristics	Baseline *n* (%)	6 Months *n* (%)
Age groups		
40–55 years	67 (44.1)	57 (37.5)
>56 years	51 (33.6)	46 (29.6)
Gender:		
Male	75 (63.6)	64 (62.7)
Female	43 (36.4)	38 (37.3)
Marital status		
Unmarried	5 (4.2)	5 (4.9)
Ever married	113 (95.8)	97 (95.1)
Education		
Primary	16 (13.6)	12 (11.7)
Secondary	67 (56.8)	58 (56.8)
Tertiary/diploma	35 (29.7)	32 (31.5)
Occupation:		
Employed in government/private	37 (31.3)	33 (32.3)
Unemployed	54 (45.8)	45 (44.2)
Self-employed	27 (22.9)	24 (23.5)
Income LKR (USD):		
<10,000 LKR (<50)	17 (14.4)	16 (13.6)
10,001–25,000 LKR (50–127)	46 (39)	41 (34.7)
>25,001 LKR (>127)	55 (46.6)	45 (38.1)
Co-morbidities:		
Diabetes only	19 (16.1)	15 (14.7)
Hypertension only	45 (38.1)	40 (39.2)
Diabetes and Hypertension	39 (33.1)	35 (34.3)
Others	15 (12.7)	12 (11.8)

LKR-Sri Lankan Rupee, USD-United States Dollars.

**Table 2 ijerph-19-05540-t002:** Anthropometric, clinical parameters, psychosocial factors and symptom burden by age group.

Characteristics	Total Mean (SD)	40–55 Years Mean (SD)	>55 Years Mean (SD)	*p*
Baseline				
Weight * (kg) ^a^	58.40 (11.3)	58.41 (12.6)	58.40 (8.9)	0.91
Height (cm) ^a^	164.01 (10.8)	163.70 (11.4)	164.42 (10.1)	0.72
BMI (kg/m^2^) ^b^	21.86 (4.5)	21.9 (4.8)	21.8 (4)	0.89
MpreBP (Hamm) ^a^	121.79 (20.4)	122.03 (20.4)	121.47 (20.6)	0.88
MpostBP (Hgmm) ^a^	127.04 (21.9)	125.26 (20.3)	129.4 (23.9)	0.31
Hb (g/dL) ^a^	9.20 (1.7)	8.93 (1.8)	9.57 (1.6)	0.05 *
S.Creatiine (mg/dL) ^a^	733.23 (329.9)	750.90 (328.5)	710 (333.7)	0.88
S.Albumin (g/dL) ^b^	3.59 (0.7)	3.59 (0.7)	3.58 (0.6)	0.50
BUN (mg/dL) ^a^	16.3 (7)	16.4 (7)	15.8 (6.5)	0.64
Stress ^a^	22.6 (6.5)	22.64 (6.3)	22.57 (6.9)	0.91
Illness perception ^a^	53.4 (9.3)	53.04 (8.3)	53.9 (10.6)	0.63
Social support ^a^	27.6 (7.3)	27.25 (7.6)	28.2 (7.09)	0.51
Symptom burden ^a^	67.04 (22.6)	66.46 (21.9)	67.8 (23.7)	0.74
6 months				
Weight * (kg) ^a^	56.86 (10.6)	57.4 (11.3)	56.1 (9.6)	0.23
Height (cm) ^a^	164.01 (10.7)	163.6 (11.5)	164.59 (9.7)	0.16
BMI (kg/m^2^) ^b^	21.33 (4.6)	27.7 (5.1)	20.9 (1.09)	0.57
MpreBP (Hgmm) ^a^	117.4 (18.4)	121.36 (19.65)	112.04 (15.7)	0.03 *
MpostBP (Hgmm) ^a^	120.1 (16.8)	122.76 (17.7)	116.8 (15.8)	0.63
Hb (g/dL) ^a^	9.45 (1.8)	9.2 (1.9)	9.8 (1.6)	0.47
S.Creatiine (mg/dL) ^a^	703.45 (253.1)	711.8 (263.4)	692.9 (241.9)	0.71
S.Albumin (g/dL) ^b^	4.05 (1.02)	4.08 (0.9)	4.01 (1.04)	0.30
BUN (mg/dL) ^a^	15.48 (5.2)	14.73 (5.3)	16.42 (5.1)	0.49
Stress ^a^	27.3 (5.1)	27.94 (4.9)	26.5 (5.3)	0.11
Illness perception ^a^	56.5 (12.3)	57.3 (11.7)	55.56 (13)	0.43
Social support ^a^	28.6 (6.2)	28.5 (5.9)	28.8 (6.7)	0.70
Symptom burden ^a^	53.1 (29.7)	55.3 (28.4)	51.1 (30.8)	0.41

a—Independent sample *t*-test, b—Mann–Whitney U test, Abbreviations: SD—Standard deviation; BMI—body mass index; MpreBP—Mean pre-dialysis blood pressure; MpostBP—Mean post-dialysis blood pressure; BUN—Blood Urea Nitrogen * *p* < 0.05.

**Table 3 ijerph-19-05540-t003:** Prevalence of symptoms by age group at baseline (*n* = 118) and 6 months (*n* = 102).

Symptom	Frequency (%)	Prevalence by Age Group *n* (%)		*p*
		40–45 Years	>55 Years	
Baseline	*n* = 118	*n* = 67	*n* = 51	
Fatigue ^b^	115 (97.5)	64 (95.5)	51 (100)	0.17
Loss/decreased libido ^b^	107 (90.7)	64 (95.5)	47 (92.2)	0.55
Loss of appetite ^b^	108 (91.5)	64 (95.5)	44 (86.3)	0.06
Swelling ^b^	109 (94.2)	61 (91)	48 (94.1)	0.39
Dry skin ^b^	109 (94.2)	61 (91)	48 (94.1)	0.38
Insomnia ^b^	107 (90.7)	61 (91)	46 (90.2)	0.56
Impotence ^b^	107 (90.7)	61 (91)	46 (90.2)	0.35
Difficulty keeping legs still ^b^	109 (94.2)	60 (89.6)	46 (90.2)	0.33
Nausea ^b^	107 (90.7)	60 (89.6)	47 (92.2)	0.44
6 months	*n* = 102	*n* = 57	*n* = 45	
Fatigue ^b^	95 (94.5)	55 (92.9)	40 (88.8)	0.49
Insomnia ^b^	94 (92.2)	52 (91.2)	42 (93.3)	0.21
Loss of appetite ^b^	92 (90.2)	52 (91.2)	40 (88.8)	0.06
Difficulty keeping legs still ^b^	92 (90.2)	51 (89.5)	41 (91.1)	0.52
Sadness ^b^	88 (86.3)	51 (89.5)	37 (82.2)	0.22
Dry skin ^b^	87 (85.3)	50 (87.7)	37 (82.2)	0.31
Changes in skin color ^b^	84 (82.4)	50 (87.7)	34 (75.6)	0.09
Lethargy ^b^	86 (84.3)	49 (86)	37 (82.2)	0.41
Joint pain ^b^	91 (89.2)	48 (84.2)	43 (95.6)	0.06
Loss/decreased libido ^b^	77 (75.5)	48 (84.2)	29 (64.4)	0.02*
Impotence	77 (75.5)	48 (84.2)	29 (64.4)	0.02*

b—Mann–Whitney U test * *p* < 0.05.

**Table 4 ijerph-19-05540-t004:** Correlation between symptom burden and socio-demographic, clinical and psychosocial factors at baseline (*n* = 118) and at 6 months (*n* = 102).

Characteristics	Baseline (*n* = 118)		6 Months (*n* = 102)	
	Mean Rank	*U*/*H*/*r* (*p*)	Mean Rank	*U*/*H*/*r* (*p*)
*Socio-demographic factors*				
Age groups: 40–55 years	58.4	1633 ^c^ (0.6)	53.7	1155 ^c^ (0.4)
>55 years	61		48.6	
Gender: Male	60.1	1585 ^c^ (0.8)	51	1198 ^c^ (0.9)
Female	59.1		51.7	
Marital status: Unmarried	70.8	226 ^c^ (0.4)	56	220 ^c^ (0.7)
Ever married	59		51.3	
Education: Up to primary	59.34	1.4 ^b^ (0.5)	45.8	1.01 ^b^ (0.6)
Up to secondary	56.6		50.6	
Tertiary	65		55.2	
Income: <10,000 (<50)	66.8	1.09 ^b^ (0.6)	55.7	0.4 ^b^ (0.8)
10,000–25000 (50–127)	59.9		51.2	
>25,000 (>127)	56.9		50.3	
Occupation: Employed	52.2	2.8 ^b^ (0.2)	52.2	2.8 ^b^ (0.2)
Self employed	61.2		61.2	
Unemployed	66		66.1	
Co-morbidities: DM	67.7	8.6 ^b^ (0.03) *	59.5	9.5 ^b^ (0.02) *
HT	48.6		40.8	
DM and HT	62.6		55.8	
Others	73.9		64.6	
*Anthropometric and clinical parameters*				
Weight (post dialysis)	-	−0.15 ^a^ (0.1)	-	−0.22 ^a^ (0.02) *
Height	-	−0.11 ^a^ (0.2)	-	−0.06 ^a^ (0.5)
BMI	-	−0.05 ^a^ (0.5)	-	−0.15 ^a^ (0.14)
MpreBP	-	0.07 ^a^ (0.5)	-	0.11 ^a^ (0.25)
MpostBP	-	0.05 ^a^ (0.6)	-	0.02 ^a^ (0.8)
Hemoglobin (mg/dl)	-	−0.17 ^a^ (0.07)	-	−0.16 ^a^ (0.9)
Creatinine (mg/dl)	-	−0.08 ^a^ (0.3)	-	−0.05 ^a^ (0.6)
Albumin (mg/dl)	-	0.03 ^a^ (0.8)	-	0.1 ^a^ (0.3)
BUN	-	0.14 ^a^ (0.1)	-	−0.06 ^a^ (0.5)
*Psychosocial factors*				
Stress	-	0.8 ^a^ (<0.005) **	-	0.34 ^a^ (<0.005) **
Illness perception	-	0.7 ^a^ (<0.005)	-	0.24 ^a^ (<0.01) *
Social support	-	−0.14 ^a^ (0.14)	-	−0.2 ^a^ (0.04) *

a—Pearson’s correlation, b—Mann–Whitney U test, c—Kruskal-Wallis H test, βe Standardized beta coefficients; U—Mann–Whitney U value; H—Kruskal-Wallis H value; r—correlation co-efficient; dependent variable: fatigue. Data in bold were significantly different from the others; BMI—body mass index; MpreBP—Mean pre-dialysis blood pressure; MpostBP—Mean post-dialysis blood pressure; BUN—Blood Urea Nitrogen; HT—Hypertension; DM—Diabetes mellitus; BUN—Blood urea nitrogen; * *p* < 0.05 **, *p* < 0.001.

**Table 5 ijerph-19-05540-t005:** Factors associated with symptom burden at baseline (*n* = 118) and at 6 months (*n* = 102).

Determinant					95% CI for β	
	USC B	SE	β	*p*	Lower Bound	Upper Bound
Baseline						
Having hypertension	1.91	2.9	0.04	0.4	−3.9	7.8
Having diabetes and hypertension	−2.75	3.3	−0.05	0.5	−9.2	3.7
Perceived stress score	2.26	0.3	0.65	<0.005 **	1.7	2.8
Illness perception score	0.5	0.21	0.21	0.02 *	0.09	0.9
6 months						
Having hypertension	−6.48	6.4	−0.11	0.3	−19.16	6.2
Having diabetes and hypertension	−7.3	7.1	−0.11	0.3	−21.16	6.63
Post dialysis body weight	−0.54	0.25	−0.19	0.03 *	−1.04	−0.04
Perceived stress score	1.7	0.53	0.28	0.003 **	0.57	2.66
Illness perception score	0.5	0.23	0.2	0.03 *	0.18	1.93
Social support	−1.05	0.44	−0.22	0.02 *	0.18	1.92

USCB—unstandardized regression coefficient; SE—standard errors of the regression coefficient; CI—confidence intervals; * *p* < 0.05 ** *p* < 0.001.

## Data Availability

The data presented in this study are available in Appendix A.

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
