# Peer review of "Factors Associated with Symptom Burden in Adults with Chronic Kidney Disease Undergoing Hemodialysis: A Prospective Study"

_ijerph, 2022, doi:10.3390/ijerph19095540_

Round 1

Reviewer 1 Report

Thank you for giving me an opportunity for reviewing interesting paper.

In this manuscript, Gunarathne and colleagues show the factors associated with symptom burden in hemodialysis patients. Authors investigated the symptom burden, socio-demographic, anthropometric measurements, clinical characteristics, stress, illness perception and social support score at the timing of baseline (beginning of the hemodialysis) and at follow-up (in 6 months). As a result, authors present factors associated with symptom burden are stress and illness perception level at baseline. In addition, high stress, illness perception level, poor social support and low body weight that is measured at post dialysis are associated with symptom burden at follow-up as predictors. While the area of research is great interest, I think this study has several methodological problems. I describe these problems and questions about this study below.

  • The authors describe “Older” patients with chronic kidney disease undergoing hemodialysis in this study. However, participants who is above 40 years old were registered. Definition of “older adults” is person aged ≥ 65 years according to WHO. Therefore, I feel authors expression is somewhat inappropriate and it is difficult to describe patients in this study “older patients”.
  • The authors describe “*** p< .001”at table legend. However, is this error for “**”?
  • The authors have to explain how you chose independent variables from all variables in multiple liner regression.
  • In Table 3, I suggest authors to add which columns indicate 40-55 years, and which is >56.
  • I wonder that there are any differences between patients aged 40-55 and >56 years about predictors associated with symptom burden in multiple regression.

I hope my comments will be helpful.

Reviewer 2 Report

  1. Please explain in detail the recruitment of patients in this prospective observational study. A flow diagram would help you to provide numbers and specific reasons for exclusion.
  2. Strength of the study is the fact that 102 out of 118 patients were evaluated in 2 different occasions (at baseline and after 6 months of follow-up). A comparison of the change in symptom burden in these patients over this period would improve this work.
  3. Please explore factors associated with the longitudinal change in symptom burden (i.e. the higher age, longer dialysis vintage, low income, stress were the major predictors of a deterioration in the reported symptoms over 6 months of follow-up).
  4. For cross-sectional correlations, please use the term “determinant” instead of using the term “predictor”.
  5. This cross-sectional study can not provide direct cause-and-effect associations and this should be clearly indicated in the limitations of the study.

Round 2

Reviewer 2 Report

No further comments.
